# The Role of Humic Acid, PP Beads, and pH with Water Backwashing in a Hybrid Water Treatment of Multichannel Alumina Microfiltration and PP Beads

**DOI:** 10.3390/membranes10010003

**Published:** 2019-12-25

**Authors:** Sungju Hwang, Yooju Lee, Jin Yong Park

**Affiliations:** Department of Environmental Sciences & Biotechnology, Hallym University, Chunchon 24252, Korea; 6402aa@naver.com (S.H.); zztm1320@naver.com (Y.L.)

**Keywords:** microfiltration, photooxidation, adsorption, polypropylene bead, hybrid treatment, water treatment, alumina

## Abstract

Photooxidation oxidizes most organic compounds by mineralizing them to small inorganic molecules. In this study, the effects of dissolved organic matter (DOM), pH, and polypropylene (PP) beads concentration on membrane fouling were investigated in a hybrid water treatment process consisting of seven-channel alumina microfiltration (pore size 1.0 μm) and pure PP beads water backwashing with UV irradiation for photooxidation. The synthetic feed was prepared with humic acid and kaolin and flowed inside the microfiltration (MF) membrane. The permeate contacted the PP beads fluidized in the gap of the membrane and module with outside UV irradiation. Membrane fouling resistance (*R_f_*) increased dramatically with an increase in the concentration of humic acid (HA) from 6 mg/L to 8 mg/L. The treatment efficiency of DOM increased dramatically, from 14.3% to 49.7%, with an increase in the concentration of HA. The *R_f_* decreased with an increase of PP beads concentration. However, maximum permeate volume (*V_T_*) was acquired at 5 g/L of PP beads. The maximal treatment efficiency of DOM was 51.3% at 40 g/L of PP beads. The *R_f_* increased with an increase in the pH of feed, and the maximum *V_T_* was acquired at a pH of 5. The maximal treatment efficiency of DOM was 52.5% at pH 9.

## 1. Introduction

The membrane separation process for separation and purification has developed dramatically during the past few decades. It can separate and concentrate all pollutants simultaneously in water by the retention of its microspores without secondary pollution and phase change. Additionally, its equipment is compact, easy to operate, and capable of continuous operation at room temperature with the advantage of low energy consumption [1]. However, membrane fouling due to the adsorption-precipitation of organic and inorganic compounds onto membranes leads to a decrease in the permeate flux, which increases membrane cleaning costs and reduces membrane life. Techniques for controlling membrane fouling remain inadequate, which is the major obstacle in the successful implementation of membrane separation technology, although considerable progress has been made in membrane fouling [2,3]. Natural organic matter (NOM) is a primary component of fouling in low-pressure membrane filtration. Various preventive measures to interfere with NOM fouling have been developed and extensively tested, such as coagulation, oxidation, ion exchange, carbon adsorption, and mineral oxide adsorption [4]. In this study, periodic water backwashing was performed to prevent the membrane fouling.

Recently, the hybrid technology of membrane separation and photooxidation by UV irradiation has effectively solved the above-mentioned membrane fouling problem [5]. Hybrid technology not only maintains the advantages of each technology, but also produces synergistic effects that overcome the limits of only one technology of membrane separation and photooxidation. In addition, pollutants such as NOM can be oxidized by UV irradiation, and organic species are rejected partially by controlling the residence time in the reacting system. In other words, the membrane is also a selective barrier for the separated molecules. Finally, hybrid technology increases photooxidation efficiency and acquires excellent effluent quality. The effect of UV-irradiation on the nano-hybrid PES-NanoZnO membrane in terms of flux and rejection efficiency has also been discussed in [6]. In addition, a comparison of the efficiency of surface water treatment in hybrid systems by coupling various advanced oxidation processes and ultrafiltration (UF) was presented by [7]. In this study, a hybrid process of ceramic membrane and pure PP beads with UV irradiation was applied for advanced water treatment.

The ceramic membranes, used in this study, have numerous advantages, which are mechanical, thermal, and chemical resistance, as well as a long lifetime. These are economically competitive as compared with polymeric membranes, because of an almost permanent lifetime [8]. Nowadays, ceramic membranes have been widely applied in water or wastewater throughout the world [9,10]. The influence of the soluble algal organic matter (AOM) characteristic on the fouling of a seven-channel tubular ceramic (MF) membrane has been investigated at lab scale by [11]. The influence of the interaction between aquatic humic substances and the algal organic AOM derived from Microcystis aeruginosa on the fouling of a ceramic MF membrane has been studied by [12].

Photooxidation has the characteristics of high efficiency, low energy consumption, and a wide range of application, by mineralizing organic compounds to small inorganic molecules and oxidization of most them, especially nonbiodegradable organic contaminants. In addition, it is an excellent technology for water pollution control. For this reason, the photooxidation technology applied in this study has been applied broadly [13,14,15,16,17,18]. In addition, degradation of humic acid (HA), which was contained in a synthetic solution used in this study, via the photoelectrocatalysis (PEC) process and the corresponding disinfection byproduct formation potential (DBPFP) were investigated; the PEC process was found to be effective in reducing dissolved organic carbon concentration [19].

In our research group, on the one hand, the results for the effect of operating conditions on a hybrid water treatment process of various ceramic membranes and titanium dioxide (TiO_2_) photocatalyst-coated polypropylene (PP) were published in “Desalination and Water Treatment” [20] and “Membr. J.” [21]. On the other hand, the roles of adsorption and photooxidation in a hybrid water treatment process of tubular carbon fiber UF and pure PP beads with UV irradiation and water backwashing were reported by our group in “Desalination and Water Treatment” [22].

In this study, the roles of humic acid, pure PP beads concentration, and pH were investigated on membrane fouling in a hybrid water treatment process of seven-channel alumina MF membranes and pure PP beads with water backwashing and UV irradiation. This research involved the unique application of pure PP beads to investigate the effect of HA, pure PP beads, and pH on the hybrid process of multichannel MF membrane as compared with our previous results [20,21,22]. Periodic water backwashing was performed for 10 s per 10 min of filtration to reduce membrane fouling. A hybrid module was composed of the MF membranes and the PP beads, which were fluidized between the gap of carbon fiber membrane and the acryl module case. The results of humic acid, pure PP beads concentration, and pH effects were compared with those of the previous study [22,23] for the hybrid process of the tubular carbon fiber UF (pore size 0.05 μm) membranes and pure PP beads with water backwashing, to investigate the effects on membrane fouling and treatment efficiency depending on membrane module types and materials.

## 2. Materials and Methods 

The seven-channel alumina MF (HC10) membranes, used in the study, were manufactured in Dongseo Industry (Seoul, Korea), with pore sizes of 0.10 μm. The specifications of the HC10 alumina membrane are listed in Table 1. In this study, 4 to 6 mm size polypropylene (PP) beads were applied and the average weight was 39.9 mg. A quantity of HA and kaolin was dissolved in distilled water, instead of natural organic matter and fine inorganic particles in a natural water source. In this experiment, it was utilized as synthetic feed water. For photooxidation, two UV lamps (F8T5BLB, Sankyo, Tokyo, Japan) irradiated UV with 352 nm from outside of the acryl module. 

To remove the turbidity and DOM, the hybrid module was packed with pure PP beads in the space between the module inside and outside of the alumina membrane. Additionally, a 100 mesh (0.150 mm) sieve, which was much smaller than 4 to 6 mm particle size of the PP beads used here, was installed at the outlet of the hybrid module to contain the PP beads in the module. 

The hybrid water treatment process (6) of multichannel MF membrane and the pure PP beads (7), which were used in the previous study [24], are shown in Figure 1 and Scheme 1. A periodic water backwashing using permeated water was performed for the multichannel MF membrane. The hybrid module (6) was supplied with PP beads fluidizing between the gap of carbon fiber membrane and the acryl module case. Then, 10 L of the synthetic water with HA and kaolin was contained in the feed tank (1). To maintain a constant viscosity of water, the temperature of the feed water was constantly maintained using a temperature circulator (3) (Model 1146, VWR, Atlanta, GA, USA). To maintain the homogeneous condition of the feed water, the synthetic feed water was continuously mixed by a stirrer (4), and it flowed into the inside of the MF membrane by a pump (2) (Procon, Standex Co., Smyna, TN, USA). A flowmeter (5) (NP-127, Tokyo Keiso, Tokyo, Japan) measured the feed flow rate. Controlling valves (9) of both the bypass pipe of the pump (2) and the concentrate pipe maintained constant flow rate and pressure of the feed water that flowed into the hybrid module. An electric balance (11) (Ohaus, Newark, NJ, USA.) measured the permeate flux treated by both the MF membrane and the PP beads. During the permeate, flux had not been measured and the permeate water flowed into the backwashing tank (13). To maintain a constant concentration of the feed water during operation, it was recycled to the feed tank (1) after the treated water was over a certain level in the backwashing tank (13). Physical washing was performed by a brush inside the tubular membrane after each operation, and the permeate flux was measured to calculate the resistances of irreversible and reversible membrane fouling. 

HA was changed from 2 mg/L to 10 mg/L, and kaolin was fixed at 30 mg/L in the experiment of HA effect. To investigate the roles of PP beads concentration and pH, kaolin and HA were fixed at 30 mg/L and 10 mg/L, respectively, in the synthetic feed water. In addition, the water backwashing time (BT) and filtration time (FT) were fixed at 10 s and 10 min, individually. During the total operation time of 180 min, the permeate flux (*J*) were measured at each condition. Transmembrane pressure (TMP) was maintained constant at 1.8 bar, the water backwashing pressure at 2.5 bar, the feed flow rate at 1.0 L/min, and the feed water temperature at 20 °C, in all experiments. The PP beads concentration was changed from 0 to 50 g/L in the module, to investigate the effect of PP beads and to investigate the pH effect, the pH was changed from 5 to 9 in the synthetic feed water.

To evaluate the treatment efficiencies of turbid materials and DOM, the quality of feed and treated water was analyzed for each experiment. To measure turbid materials and DOM, turbidity was measured using a turbidimeter (2100N, Hach, Ames, IA, USA) and UV_254_ absorbance was analyzed using a UV spectrophotometer (Genesys 10 UV, Thermo, Pittsburgh, PA, USA). The detection limits of turbidimeter and UV spectrophotometer were 0 to 4000 NTU (±0.001 NTU) and −0.1 to 3.0 cm^−1^ (±0.001 cm^−1^) correspondingly. Before measuring UV_254_ absorbance, each sample was filtered using a 0.2 μm syringe filter to remove turbid materials. 

To clean the ceramic membrane and apparatus after each experiment was finished, all of the synthetic solution was discharged from the hybrid water treatment system, and distilled water was circulated through the line of the system for 15 min. To combust fouling materials inside the membrane, the PP beads and the membrane were separated from the module, and the membrane was ignited at 550 °C in a furnace for 30 min. To wash out organic or inorganic pollutants after cooling the membrane, the membrane was immersed in a nitric acid (HNO_3_) of 15% for 24 h, and in a sodium hydroxide (NaOH) solution of 0.25 N for 3 h, and kept in distilled water for 24 h for rinsing. Before beginning a new experiment, the water permeated flux (*J_w_*) was measured to check the membrane recovery when a normal operation was performed with distilled water, after the recovered membrane was installed inside the module. The recovered membrane was used in all of the experiments to reduce the influence of membrane condition on the treatment efficiency.

## 3. Results and Discussions

The effects of HA, pure PP beads concentration, and pH were investigated in the hybrid water treatment process of seven-channel alumina MF (HC10) membrane and pure PP beads with periodic water backwashing and UV irradiation. Membrane resistance, boundary layer, and membrane fouling (*R_m_*, *R_b_*, and *R_f_*) were calculated from permeate flux (*J*) data using the resistance-in-series filtration equation (*J* = Δ*P*/(*R_m_* + *R_b_* + *R_f_*)) which was the same method as the previous study [24], where Δ*P* is the transmembrane pressure. For a new membrane, the equation was simplified to *J* = Δ*P*/*R_m_* because there were no boundary layer resistances, and membrane fouling and *R_m_* could be determined using *J* data for a new membrane. For the synthetic water of HA and kaolin, the equation was modified to *J* = Δ*P*/(*R_m_* + *R_b_*) at initial time, and *R_b_* could be determined using *J*_0_ and *R_m_* data. Irreversible and reversible membrane fouling resistances (*R_if_*, *R_rf_*) could be found from *J* data after physical washing using a brush inside the membrane.

### 3.1. Effect of HA Concentration on Membrane Fouling and Treatment Efficiency

The membrane fouling resistance (*R_f_*) was significantly influenced by humic acid (HA), which was one of the major natural organic matters (NOM) in lakes or rivers, and *R_f_* was the minimum at HA 0 g/L, and almost constant at 2 to 6 mg/L of HA; however, it increased dramatically with an increase in the concentration of HA from 6 mg/L to 8 mg/L, and then, decreased suddenly in a concentration of HA from 8 mg/L to 10 mg/L , as shown in Figure 2a. This means that DOM, similar to HA, could drive membrane fouling more severely on the surface and inside the alumina membrane, with an increase of HA concentration from 6 mg/L to 8 mg/L in water. However, the membrane fouling decreased from 8 mg/L to 10 mg/L of HA, because the thick fouling cake on the membrane could be removed by water backwashing at 10 mg/L of HA. As summarized in Table 2, the membrane resistance (*R_m_*) was controlled at a constant value by combustion in a furnace and washing with an acid and alkali solution. The final *R_f_* (*R_f_*_,180_) value after operating for 180 min at HA 8 mg/L was 1.963 × 10^9^ kg/m^2^s, which was 3.24 times higher than 0.606 × 10^9^ kg/m^2^s of the *R_f_*_,180_ value at HA 0 mg/L. 

In the previous study [22] for the hybrid water treatment process of the same tubular carbon fiber UF and the same pure PP beads, *R_f_* increased dramatically with an increase in the concentration of HA from 2 mg/L to 10 mg/L. The increasing rate of *R_f_* was very high, when HA increased from 6 mg/L to 10 mg/L, in particular. This means that DOM, similar to HA, could drive membrane fouling more severely on the surface and inside the carbon fiber membrane, with an increase of HA concentration in water, and specifically at a high HA concentration. There was a little different trend as compared with the study of HC10, depending on the membrane type and material. The *R_f_*_,180_ value after operating for 180 min at HA 10 mg/L was 6.998 × 10^9^ kg/m^2^s, which was 3.94 times higher than 1.775 × 10^9^ kg/m^2^s of the *R_f_*_,180_ value at HA 2 mg/L. The *R_f_*_,180_ value was much higher than those of this result of HC10 in this study, meaning that membrane fouling could be much less developed in the seven-channel HC10 membrane as compared with the tubular C005 membrane.

As shown in Figure 2b, the dimensionless permeate flux (*J*/*J*_0_), where *J*_0_ was the initial permeate flux predicted using the initial two data by an extrapolation method, was plotted according to HA concentration. The *J*/*J*_0_ overlapped in the range of HA 2 to 6 mg/L, and it showed the highest values at HA 0 mg/L and the lowest at HA 8 mg/L during 180 min; however, it was much higher at HA 10 mg/L. This means that the permeate flux could be maintained low at HA 8 mg/L and increased at HA 10 mg/L, because the membrane fouling was developed severely by DOM, and the cake layer could be removed at HA 10 mg/L by water backwashing. 

As arranged in Table 2, the final *J*/*J*_0_ after operating for 180 min (*J*_180_/*J*_0_) was 0.578 at HA 0 mg/L, which was 1.88 times higher than 0.307 at HA 8 mg/L. In the previous work [22] of the C005, *J*/*J*_0_ tended to decrease with an increase in the HA concentration from 2 to 10 mg/L, specifically from 6 to 8 mg/L, because of the membrane fouling development by the more DOM. The *J*_180_/*J*_0_ was 0.241 at HA 2 mg/L, which was 2.84 times higher than 0.085 at HA 10 mg/L. This means that the permeate flux decline rate was affected more severely by HA concentration in the hybrid process of C005 than HC10, because the membrane fouling increased dramatically with an increase in HA concentration. 

In addition, the permeate volume (*V_T_*) of 14.18 L at HA 0 mg/L was 1.57 times higher than 9.05 L of *V_T_* at HA 8 mg/L, as shown in Table 2, because high flux was maintained at HA 4 mg/L during 180 min as compared in Figure 2b. In [22], the result of the tubular carbon fiber C005, *V_T_* at HA 2 mg/L was 4.98 L, which was 2.59 times higher than 1.92 L of *V_T_* at HA 10 mg/L. The difference rate of *V_T_* was higher in the hybrid process of C005 than HC10. This means that DOM affected membrane fouling more at tubular UF C005 membrane than multichannel MF HC10. 

As listed in Table 3, the treatment efficiency of turbidity showed an increasing trend, with an increase of HA concentration. Furthermore, in the result by [22] with C005, the treatment efficiency of turbidity was almost constant. This means that the DOM could affect the treatment of suspended particles, such as kaolin, in the hybrid process of the multichannel alumina MF and the pure PP beads, because the thick membrane fouling layer rejected kaolin particles at a high HA concentration. However, DOM could not affect the process of tubular carbon fiber UF membrane, because of the tubular membrane type.

As shown in Table 4, the treatment efficiency of UV_254_ absorbance, which means the concentration of DOM, increased with an increase in HA concentration, and finally showed the maximum 49.7% at HA 10 mg/L. This means that DOM could be treated more effectively at a high DOM condition in the hybrid water treatment process of seven-channel alumina MF and pure PP beads. The maximum treatment efficiency of DOM was 69.3% at HA 6 mg/L, in the previous work [22] for the hybrid process of the tubular carbon fiber MF membrane and the same PP beads. From 2 mg/L to 6 mg/L of HA, most of DOM could be adsorbed on fouling materials inside the membrane or retained by cake layer on the membrane, and the remained things passed through the membrane adsorbed or oxidized by the PP beads and UV. It happened that the treated water quality of HA increased less slowly than the feed water quality. However, above 6 mg/L of HA, most of DOM passed through the membrane and could not be treated by adsorption or photooxidation by the PP beads and UV. And the treated water quality of HA increased more rapidly than the feed water quality.

### 3.2. Effect of Pure PP Beads on Membrane Fouling and Treatment Efficiency

In this study, the pure PP beads were induced for adsorption of DOM and turbid matters. The effect of pure PP beads concentration was investigated at the most severe HA 10 mg/L condition, at kaolin 30 mg/L, and pH 7. The membrane fouling resistances (*R_f_*) showed the highest at 50 g/L and the lowest at 5 g/L during the 180 min operation, as shown in Figure 3a. This means that the optimal PP beads concentration could be 5 g/L to control the membrane fouling and high permeate flux in this hybrid process of 7-channel alumina MF HC10 and PP beads. 

In the previous work by [23] for the hybrid process of the tubular carbon fiber UF C005 and the same PP beads, the *R_f_* showed the highest at 50 g/L during the 180 min operation, and the lowest at 5 g/L until 120 min, and at 0 g/L of PP beads after 120 min. This result almost agreeds with the trend of PP beads effect in this study and showed that the effect of PP beads concentration on membrane fouling could not depend on the membrane type and materials in this hybrid water treatment process of ceramic membrane and pure PP beads.

In the hybrid process of ceramic membrane and PP beads, the boundary layer resistance (*R_b_*), which was produced by concentration polarization on the membrane surface, was the lowest at 50 g/L of PP beads independent of membrane type and materials, as shown in Table 5. This means that the frequent colliding of more PP beads on the membrane surface could reduce the concentration polarization for both HC10 and C005. The *R_f_*_,180_ after 180 min was the highest, i.e., 12.94 × 10^9^ kg/m^2^s, at 50 g/L, which was 3.06 times higher than 4.23 × 10^9^ kg/m^2^s for the *R_f_*_,180_ at 0 g/L of PP beads. The *R_rf_* showed an increasing trend, with an increasing PP beads concentration from 0 g/L to 50 g/L; however, the minimum *R_if_* was at 40 g/L and the maximum was at 0 g/L of PP beads. This means that the reversible membrane fouling could be inhibited at 50 g/L of PP beads, because the optimal amount of PP beads captured the turbid or organic materials by adsorption.

In the previous result by [23] for the hybrid process of the C005 and the pure PP beads, the *R_f_*_,180_ after 180 min was the highest of 4.892 × 10^9^ kg/m^2^s at 50 g/L, which was 1.48 times higher than 3.306 × 10^9^ kg/m^2^s of the *R_f_*_,180_ at 0 g/L of PP beads. These *R_f_*_,180_ were much higher than those of HC10 in this study; however, the trends depending on PP beads concentration was exactly the same in this hybrid process. Additionally, the *R_rf_* increased with an increasing PP beads concentration from 0 g/L to 50 g/L of PP beads; however, the *R_if_* was the minimum at 50 g/L and the maximum at 10 g/L of PP beads. This result was also almost in agreement with that of this study using HC10.

The *J*/*J*_0_ was compared to investigate the effect of PP beads on the relative decline of permeate flux, as shown in Figure 3b. The *J*/*J*_0_ maintained higher until 90 min at PP beads 5 g/L than those at other PP beads concentration, and showed the lowest at PP beads 50 g/L after 60 min. As arranged in Table 5, the *J*_0_ and *J*_180_ decreased to 40 g/L and to 50 g/L, respectively, with an increase in PP beads concentration, because the *R_b_* and *R_f_* increased to 40 g/L and to 50 g/L of PP beads, correspondingly. Finally, the *J*_180_/*J*_0_ after 180 min of operation at 0 g/L of the PP beads was the maximum 0.230, which was 2.13 times higher than 0.108 at 50 g/L. However, the *V_T_* was the highest, 11.75 L at 5 g/L of PP beads, because *J* maintained higher all through the operation than those of other PP beads conditions. 

In the previous work by [23] for the hybrid process of the C005 and the pure PP beads, the *J*/*J*_0_ showed higher during the 5 to 90 min at 5 g/L than those at other PP beads concentration. The *J*_0_ decreased and *J*_180_ increased as increasing PP beads concentration, because the *R_b_* was the minimum at 50 g/L of PP beads. In addition, the *J*_180_/*J*_0_ after 180 min of operation at 0 g/L of the PP beads was the maximum 0.126, which was 1.62 times higher than 0.078 at 50 g/L. The increasing rate, 1.62 of *J*_180_/*J*_0_ for C005, was much lower than 2.13 for HC10, because PP beads controlled more effectively the membrane fouling in the hybrid process of HC10 than C005. Also, the *V_T_* was the highest 4.99 L at 5 g/L of PP beads for the C005 process, which was the exact same trend of HC10 in this study.

As arranged in Table 6, the treatment efficiencies of turbidity were almost constant in the range of 97.5% and 98.9% in the hybrid process of HC10 MF, independent of the pure PP beads concentration. This means that the tubid matters could be treated effectively, independent of PP beads concentration in this hybrid process. In the previous result by [23] for the hybrid process of the C005 and the pure PP beads, the treatment efficiencies of turbidity were the highest, i.e., 99.3% at 30 g/L. This means that the optimal PP beads concentration could be 30 g/L to treat the turbid matter in the hybrid process of C005 UF membrane.

As shown in Table 7, the treatment efficiency of DOM (UV_254_ absorbance) did not show a trend; however, that was the maximal of 51.3% at 5 g/L of PP beads. This proves that the optimal PP beads concentration was 5 g/L to remove DOM in this hybrid process of HC10 MF and pure PP beads. In the previous work by [23] for the hybrid process of the C005 and the pure PP beads, the treatment efficiency of DOM showed a trend to increase dramatically from 75.9% to 84.1%, with an increase in the PP beads concentration. It was much higher than those for the hybrid HC10 MF process in this study, because the secondary gel layer could be formed denser on the UF membrane surface of smaller pore size than MF. Finally, the more PP beads could adsorb the more efficiently DOM on the surface of PP beads in the hybrid C005 UF process.

### 3.3. Effect of pH on Membrane Fouling and Treatment Efficiency

To investigate the pH effect on membrane fouling and treatment efficiency, the pH of synthetic feed water was controlled by nitric acid (HNO_3_) and sodium hydroxide (NaOH). As shown in Figure 4a, the *R_f_* maintained the highest values at pH 9, and the lowest at pH 5 during 180 min. There was a dramatic trend to increase as the pH increased from 5 to 9 in the hybrid process of HC10 MF. As arranged in Table 8, the *R_f_*_,180_ and R_rf_ were the minimum at pH 5; however, the *R_b_* was the lowest at pH 6. Conclusively, the *R_f_*_,180_ increased dramatically with an increase in pH, and was the highest, 20.20 × 10^9^ kg/m^2^s, at pH 9, which was 4.58 times higher than 4.41 × 10^9^ kg/m^2^s at pH 6. The *R_rf_* and *R_if_* were the maximum at pH 9 and pH 8, respectively. This means that the reversible and irreversible membrane fouling, and concentration polarization could be inhibited at acid condition, because both the membrane and humic materials had a negative surface charge at acid conditions below a pH of 7, as reported that the surface charge of ZrO_2_ membrane was changed depending on the pH [25]. The surface charge of seven-channel alumina HC10 membrane, used in this study, could be changed depending on the pH, because those were the similar ceramic membranes as ZrO_2_ membrane.

In the previous result by [23] for the hybrid process of the C005 and the pure PP beads, the *R_f_* showed the highest at a pH of 9 after 90 min and the lowest at a pH of 5 after 150 min, and finally have a trend to increase, with an increase in pH from 5 to 9. This result agreeds exactly with that in the hybrid process of HC10 in this study. The *R_b_* and *R_if_* were the minimum at a pH of 5; however, the *R_f_*_,180_ and *R_rf_* were the lowest at a pH of 6. However, this trend did not match the result for HC10 in this study. Conclusively, the *R_f_*_,180_ was the highest, 5.51 × 10^9^ kg/m^2^s, at a pH of 9, which was 1.32 times higher than 4.16 × 10^9^ kg/m^2^s at a pH of 6. The *R_rf_* and *R_if_* were the maximum at pH 9. The increasing rate, i.e., 1.32 of *R_f_*_,180_ for C005 was much lower than 4.58 for HC10, because pH controlled more effectively the membrane fouling in the hybrid process of HC10 than C005.

As compared in Figure 4b to investigate the pH effect on relative permeate flux, the *J*/*J*_0_ showed a trend to decrease dramatically, with an increase in the pH from 5 to 9. The *J*_180_/*J*_0_ after 180 min of operation at pH 5 was 0.306, which was 2.94 times higher than 0.104 at pH 9, as shown in Table 8. And the *J*_180_ was the maximum 327 L/m^2^h at pH 6, and the minimum 270 L/m^2^h at pH 9. This means that the high permeate flux could be acquired at pH 5, because the membrane fouling was inhibited by repulsion force between the carbon fiber membrane and humic materials, which had the same negative surface charge, as reported that the surface charge of ZrO_2_ membrane was changed depending on the pH [25]. Finally, the *V_T_* was the highest of 8.12 L at pH 5, because the *J* could maintain highly during a 180 min operation. 

In the previous work by [23] for the hybrid process of the C005 and the pure PP beads, the *J*_180_/*J*_0_ after 180 min of operation at a pH of 6 was 0.099, which was 1.46 times higher than 0.068 at a pH of 9. The increasing rate, 1.42 of *J*_180_/*J*_0_ for C005, was much lower than 2.94 for HC10, because the pH maintained higher the permeate flux in the hybrid process of HC10 than C005. Additionally, J_180_ was the maximum 138 L/m^2^h at a pH of 6, and the minimum, 107 L/m^2^h, at a pH of 9. Finally, the *V_T_* was the highest of 3.85 L at pH 8, because the permeate flux could maintain highly during operation. These trends did not match exactly with the result for HC10 in this study, because of the different membrane type and material.

As arranged in Table 9, the treatment efficiency of turbidity decreased slightly from 98.6% to 98.0%, with an increase in pH from 5 to 9. This means that the turbid matter, such as kaolin, could be removed more effectively at acidic pH conditions in the hybrid process of the alumina HC10 MF membrane with periodic water backwashing. Y. Zhao et al. [26] reported that zeta-potential of alumina membrane decreased with an increase in pH. This phenomenon could be explained that the treatment efficiency of turbidity decreased slightly, because the decrease of zeta potential of HC10 membrane resulted in a decrease of the electroviscous effect. However, in the previous result [23] for the hybrid process of the C005 and the pure PP beads, the treatment efficiency of turbidity was almost constantly above 98.7%, independent of the pH. This means that the pH could not affect treating the turbid matter for carbon fiber UF process with water backwashing.

As shown in Table 10, the treatment efficiency of DOM showed the highest value at a pH of 9. This means that DOM could be removed effectively at alkalic condition, because of the secondary layer on the membrane surface accumulated by the most severe membrane fouling. However, in the previous result [23] for the hybrid process of the C005 and the pure PP beads, the treatment efficiency of DOM was the highest at a pH of 5. This means that DOM could be removed effectively at acid condition, because of repulsion force between the carbon fiber membrane and humic materials.

## 4. Conclusions

In this study, the roles of humic acid, PP bead concentration, and pH were investigated on membrane fouling and treatment efficiency of turbid matter or DOM in the hybrid process of the multichannel alumina MF membrane and pure PP beads with periodic water backwashing. The results of pH effect were compared with those of the previous study [22,23] in the hybrid process of the tubular carbon fiber UF and the same PP beads with water backwashing. In conclusion, the following results could be extracted from these investigations.
(1)DOM, such as HA, could drive membrane fouling more severely on the surface and inside the alumina membrane, with an increasing HA concentration in water; however, the thick fouling cake on the membrane could be removed by water backwashing at 10 mg/L of HA. DOM could affect the treatment of suspended particles, such as kaolin, in the hybrid process of the multichannel alumina MF and the pure PP beads; however, it could not affect the process of tubular carbon fiber UF membrane. DOM could be treated more effectively at high DOM condition in the hybrid water treatment process of seven-channel alumina MF and pure PP beads.(2)The optimal PP beads concentration could be 5 g/L to control the membrane fouling in this hybrid process of seven-channel alumina MF HC10 and PP beads. The tubid matters could be treated effectively, independent of PP beads concentration in this hybrid process. The optimal PP beads concentration was 5 g/L to remove DOM in this hybrid process. The optimal PP beads concentration to reduce the turbid matter could be 30 g/L in the hybrid process of C005 UF membrane.(3)The reversible and irreversible membrane fouling, and concentration polarization could be inhibited at acid condition, because both the membrane and humic materials had a negative surface charge at acid conditions below pH 7. The turbid matter could be removed more effectively at an acidic pH condition in the hybrid process of the alumina HC10 MF membrane; however, pH could not affect treating the turbid matter for carbon fiber UF process. DOM could be removed effectively in alkalic condition, because of the secondary layer on the membrane surface accumulated by the most severe membrane fouling.

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
