# Peer review of "The Role of Humic Acid, PP Beads, and pH with Water Backwashing in a Hybrid Water Treatment of Multichannel Alumina Microfiltration and PP Beads"

_membranes, 2019, doi:10.3390/membranes10010003_

Round 1

Reviewer 1 Report

Comments:

In this manuscript, the authors have investigated the effect of concentration of dissolved organic matters (DOM) and polypropylene beads as well as feed pH on the fouling of 7 channel alumina MF membrane. The membrane is tested with humic acid as foulant under the operating conditions in this study. The results are compared to previous reports. In my opinion, the manuscript can be published in Membranes after major revision.

1.      The authors stated in introduction “ceramic membranes used in this study maintain competitive prices compared with polymer membrane, and have a lot of advantages, which are mechanical, thermal and chemical resistance, and long life time.” Please cite the most relevant reference for this statement.

2.      Authors appear to rely heavily on resistance-in-series model to explain the fouling behavior in this study. A short introduction about the model should be included in the manuscript. Maybe at the start of results and discussion.

3.      The authors mentioned the decrease in Rf value on the increase in humic acid concentration from 8mg/L to 10 mg/L which is also represented in Figure 2a (which is inverse of the phenomenon observed when humic acid concentration was increased from 2 to 8 mg/L). Also, the normalized flux plotted in figure 2b has the lowest value for 8mg/L humic acid feed compared to that of 10 mg/L humic acid. This is contradictory to the resistance-in-series model. The authors should explain this anomaly.

4.      Comment on how increasing the concentration of HA affected the treatment efficiency of turbidity would be helpful for readers.

5.      A slight decrease in treatment efficiency of turbidity was observed on increasing the pH from 5 to 9 (Table 9). A brief comment on this observation would be helpful. How the acidic pH assist in the efficient removal of Kaolin in the hybrid process with alumina HC10 MF membrane?

6.      No data on molecular weight cutoff for the membranes is reported. A comment on MWCO would be helpful for readers.

Other recommendations:

1.      Some of the sentences are too long and needs to be read repeatedly to understand them in one reading. Please consider to rephrase them into smaller sentences. Example: starting sentence of abstract (line 10-14), line (63-67).

2.      Why would authors use Nitric acid for adjusting pH? Normally, more common salt such as HCl should be used.

3.      If possible microscopy images on the membrane surface before and after fouling should be included in the manuscript.

4.      Figure 4 appears to be vertically stretched. The authors should provide figures with clear axis labels.

5.      Grammatical errors are found in the manuscript. Recommend thorough editing of the manuscript.

Author Response

Thanks a lot for your kind and detail comments. Answering each comment and revising the article as a attached file.

Reviewer 2 Report

In this work, the roles of humic acid, PP bead concentration and pH were investigated on membrane fouling and treatment efficiency in turbid matter and DOM in the hybrid process of the multi-channels alumina MF membrane and pure PP beads with periodic water back-washing. HA and kaolin were dissolved in distilled water, which was used as the feed water. The authors also compared the previous results of the C005 and the pure PP beads with those in this study. Here are some suggestions for improving the quality of this manuscript:

The highlights of this work should be clearified. The authors have published some papers such as “Roles of polypropylene beads and photo-oxidation in hybrid water treatment process of alumina MF and photocatalyst-coated PP beads”,“Effect of water back-flushing time and polypropylene beads in hybrid water treatment process of photocatalyst-coated PP beads and alumina microfiltration membrane” and “Roles of polypropylene beads and pH in hybrid water treatment of carbon fiber membrane and PP beads with water back-flushing”. What is the difference and key innovation of this work?

2.The influence of humic acid, PP bead concentration and pH on membrane performance is not identified, especially for humic acid and PP bead concentration. And it is necessary to explain the specific reasons.

Although the method was citied in the previous study, I think it would be better to explain more details as well as formula of Rf ,Rm and other relevant parameters.

4.In 3.1, “Effect of HA concentration on membrane fouling and treatment efficiency”, it would be better if the authors provide the concentration of HA 0mg/L.

5.It would be suggested that the details of all the figures keep consistent, such as the location of legend and the size of all the font.

6.It will be more intuitive for us to monitor membrane fouling if there are SEM images or some in situ monitoring results.

For feed water and treated water, under different parameters such as humic acid, PP bead concentration and pH, the water quality and treatment efficiency of turbidity and NOM can be compared in a figure in order to present the results more clearly. There are some errors in the manuscript, such as spelling and numbering. For instance, “dending on” in page 5 and “effedt” in page 10. I think it is necessary to carefully check the whole manuscript.

Author Response

Thanks a lot for your kind and detail comments. Answering each comments and revising the article as attached file.

Round 2

Reviewer 1 Report

Satisfied with the revision made by authors. Please accept the manuscript in its revised format.

Author Response

Thanks a lot for your accepting our revsied article.

Reviewer 2 Report

The authors have done some efforts to improve the quality of the manuscript. However, some comments or suggestions are still not well considered, such as Comments-4, 6. Otherwise, I would like to reject it

Author Response

Thanks a lot for your detail review and comments.

We have done an extra experiment at HA 0 mg/L, and the results were added in this article.

However, because we have only one of HC10 membrane because the manufacture company was closed now by economically problem, it is so sorry not to take the SEM picture of membrane surface by cutting out the membrane.

Please understand our limited situation for HC10 membrane.

Round 3

Reviewer 2 Report

It can be accepted